# Atelocollagen exhibits superior performance compared to growth factors in upregulating proteins associated with tendon healing

Sung-Jin Park[1], Jong Pil Yoon[2], Kyung-Soo Oh[1], Seok Won Chung[1]*

**1** Department of Orthopedic Surgery, Center for Shoulder and Elbow Surgery, Konkuk University School of Medicine, Seoul, Korea, **2** Department of Orthopaedic Surgery, School of Medicine, Kyungpook National University, Daegu, Korea

\* smilecsw@gmail.com

## Abstract

### Purpose

We aimed to compare the effects of atelocollagen (AC) and individual growth factors on the expression of key molecular markers associated with tendon healing.

### Methods

C2C12 myoblasts were cultured in Dulbecco's Modified Eagle Medium (DMEM) containing 5% fetal bovine serum (FBS) and treated with 1 nM or 10 nM of Atelocollagen (AC), bone morphogenetic protein-2 (BMP-2), transforming growth factor-beta 1 (TGF-β1), insulin-like growth factor-1 (IGF-1), or vascular endothelial growth factor (VEGF) for 5 days. After 5 days of treatment, cells were harvested from the culture medium, and Western blot analysis was performed to quantify the expression of phosphorylated extracellular signal-regulated kinase (p-ERK), Collagen type I (Col I), Collagen type III (Col III), and Tenascin C (TnC). Additionally, immunofluorescence staining was conducted to qualitatively assess cytoskeletal organization and cell adhesion, which are key factors in tendon healing.

### Results

In the AC-treated groups, the expression levels of p-ERK, Col I/Col III, and TnC were significantly higher compared to the groups treated with individual growth factors (BMP-2, IGF, VEGF, and TGF-β1) ($p < 0.05$). These changes were dose-dependent, as there was no significant difference in protein expression between AC and the growth factors at 1 nM, whereas at 10 nM, AC treatment resulted in a significant increase ($p < 0.05$). In the cell proliferation assay, C2C12 myoblasts treated with AC at 10 nM exhibited significantly higher proliferation rates compared to those treated with individual growth factors ($p < 0.05$). Additionally, immunofluorescence analysis

**Data availability statement:** All relevant data are within the paper and its Supporting Information files.

**Funding:** "This work was supported by the National Research Foundation of Korea (NRF) grant funded by the Korea government (No. RS-2023-00249219)." The funders had no role in study design, data collection and analysis, decision to publish, or preparation of the manuscript.

**Competing interests:** The authors have declared that no competing interests exist.

revealed greater cytoskeletal alignment in AC-treated cells, suggesting enhanced cell adhesion, structural organization, and mechanical stability.

## Conclusions

AC significantly upregulated key molecular markers involved in cell proliferation, extracellular matrix remodeling, and tendon structural integrity more effectively than individual growth factors. The increased expression of these genes in myoblasts suggests AC's potential role in promoting tendon healing.

## Clinical relevance

Through the modulation of key molecular pathways critical to tendon healing, AC presents strong potential as an effective biological augmentation strategy for improving tendon-to-bone interface healing after surgical repair.

---

## Introduction

Tendon-to-bone interface (TBI) healing remains a major clinical challenge, as surgical repair alone is often insufficient to fully restore the structural and biomechanical properties of the native tendon-bone junction [1]. The intrinsic regenerative capacity of tendons is limited due to their hypovascular nature and low cellularity, resulting in scar-mediated healing rather than functional tissue regeneration [1,2]. This process leads to disorganized collagen deposition, inadequate mechanical strength, and incomplete restoration of the fibrocartilaginous transition zone, which are associated with high rates of repair failure, functional deficits, and an increased risk of re-injury [3,4]. Additionally, excessive scar tissue formation disrupts native collagen alignment, contributing to chronic stiffness, muscle atrophy, and joint instability, which ultimately impairs mobility, strength, and quality of life [4,5].

Efforts to enhance tendon healing have increasingly focused on biological augmentation strategies aimed at improving tissue regeneration at the molecular level [5,6]. Since natural tendon repair is often compromised by excessive scar formation and insufficient extracellular matrix (ECM) remodeling, biologically driven approaches seek to support cellular activity, regulate inflammatory responses, and promote organized collagen deposition [7]. Among these strategies, growth factor-based therapies have been widely studied, with bone morphogenetic protein-2 (BMP-2), transforming growth factor-beta 1 (TGF-β1), insulin-like growth factor-1 (IGF-1), and vascular endothelial growth factor (VEGF) demonstrating the ability to stimulate ECM remodeling, enhance collagen synthesis, and promote tendon cell proliferation [8,9]. However, their short biological half-life, rapid degradation, and challenges in controlled delivery have limited their clinical application [8,10,11].

While growth factor-based therapies have demonstrated the ability to stimulate ECM remodeling and promote tendon cell proliferation, their clinical application remains limited due to rapid degradation, short biological half-life, and challenges in sustained delivery [12]. As an alternative, Atelocollagen (AC) has emerged as a promising biomaterial for tendon repair, offering a more stable and controlled approach to enhancing

tendon regeneration [13]. Unlike exogenous growth factors, which require repeated administration or carrier systems due to their rapid degradation and short biological half-life, AC possesses inherent structural stability and a sustained bioactive profile, allowing it to continuously promote cellular activity without the need for additional delivery carriers [13].

This study aimed to compare the effects of AC and growth factors on the regulation of tendon healing-related gene markers. We hypothesized that AC would be as effective as, or more effective than, individual growth factors in upregulating the expression of key genes involved in tendon healing. To evaluate AC's potential as an alternative or complementary strategy to growth factor-based therapies, we analyzed the expression of molecular markers in vitro.

## Methods

### Cell proliferation measurement

C2C12 muscle cells were obtained from Korean Collection for Type Cultures (KCTC, Cat. No. KCTC-10410, Jeongeup, Korea). The cells were cultured in Dulbecco's Modified Eagle's Medium (DMEM; DMEM; Gibco, Thermo Fisher Scientific, Waltham, MA, USA) containing 10% fetal bovine serum (FBS; Gibco, Thermo Fisher Scientific) and 1% penicillin-streptomycin (PS) at 37°C, 5% $CO_2$. When the cells reached a confluent density of approximately 70–80%, they were detached from the attachment surface using 0.05% trypsin-0.53 mM EDTA solution and then collected by centrifugation. The collected cells were re-cultured on a new culture dish, and the culture medium was changed every 2–3 days to maintain an optimal cell growth environment.

Cell proliferation was assessed using the MTT (3-(4,5-dimethylthiazol-2-yl)-2,5-diphenyltetrazolium bromide) assay [14]. C2C12 cells were seeded into 96-well plates at a density of $5 \times 10^4$ cells/well and incubated in DMEM supplemented with 10% FBS at 37°C and 5% $CO_2$. After 24 hours, cells were treated with AC (1nM and 10nM) or individual growth factors (BMP-2, TGF-β1, IGF-1, VEGF, each at 1nM and 10nM), with an untreated control group maintained under the same conditions. After 6 days of treatment, 10 μL of MTT reagent (5 mg/mL in PBS) was added to each well, and the plates were incubated for 4 hours at 37°C. The culture medium was then removed, and 150 μL of dimethyl sulfoxide (DMSO) was added to dissolve the formazan crystals. Absorbance was measured at 570 nm using a microplate reader, with background correction at 630 nm.

### Western blot analysis

Western blot analysis was performed to evaluate the expression of key signaling, muscle differentiation, ECM, tendon, adipogenic, and mitochondrial proteins. C2C12 cells were cultured and treated with Atelocollagen (AC, 1nM and 10nM) or individual growth factors (BMP-2, TGF-β1, IGF-1, VEGF, each at 1nM and 10nM) for 6 days [15–18]. After treatment, cells were harvested and lysed using RIPA buffer supplemented with protease and phosphatase inhibitors. The lysates were centrifuged at 12,000×g for 10 minutes at 4°C, and the supernatants were collected for protein quantification using the bicinchoninic acid protein assay. Equal amounts of protein (20–30 μg) were loaded onto SDS-PAGE gels (10–12%) and separated by electrophoresis at 100V for 90 minutes. The proteins were transferred to PVDF membranes using a wet transfer system (250 mA, 2 hours at 4°C). Membranes were blocked with 5% BSA or non-fat dry milk in TBST (Tris-buffered saline with 0.1% Tween-20) for 1 hour at room temperature. Membranes were incubated overnight at 4°C with primary antibodies against p-ERK (Thr202/Tyr204; rabbit monoclonal, Cell Signaling Technology, Cat. No. 9101, Danvers, MA, USA), ERK1/2 (rabbit monoclonal, Cell Signaling Technology, Cat. No. 9102), Collagen type I (rabbit polyclonal, Abcam, Cat. No. ab34710, Cambridge, UK), Collagen type III (rabbit polyclonal, Abcam, Cat. No. ab7778), TnC (abbit polyclonal, Sigma-Aldrich, Cat. No. T2551, St. Louis, MO, USA), and GAPDH (mouse monoclonal, Santa Cruz Biotechnology, Cat. No. sc-32233, Dallas, TX, USA). Membranes were washed three times with TBST and incubated with HRP-conjugated secondary antibodies for 1 hour at room temperature. The signals were visualized using an enhanced chemiluminescent substrate and captured with a Western blot imaging system. Densitometry analysis was performed using ImageJ software, and relative protein expression levels were normalized to GAPDH. All experiments were independently repeated three times.

## Immunofluorescence

This experiment aimed to evaluate cytoskeletal organization and actin filament distribution in C2C12 cells following treatment with AC and growth factors (VEGF and TGF-β1) using immunofluorescence staining. After treatment with AC, VEGF, or TGF-β1 at 10 nM, C2C12 cells were washed twice with phosphate-buffered saline (PBS) and fixed with 4% paraformaldehyde for 15 minutes at room temperature. Cells were then permeabilized with 0.1% Triton X-100 in PBS for 10 minutes, followed by three PBS washes. Nonspecific binding was blocked by incubating cells with 5% bovine serum albumin (BSA) in PBS for 1 hour at room temperature. Primary antibodies targeting β-actin (cytoskeletal marker, red fluorescence) and DAPI (nuclear stain, blue fluorescence) were applied overnight at 4°C in 1% BSA in PBS. Cells were washed three times with PBS and incubated with fluorescently labeled secondary antibodies (Alexa Fluor 594 for actin and DAPI for nuclei) for 1 hour at room temperature in the dark. After a final PBS wash, coverslips were mounted using Fluoromount-G, and fluorescence images were captured using a confocal microscope under standardized exposure conditions. Fluorescence intensity was quantified using ImageJ software, with actin expression normalized to nuclear staining. All experiments were independently repeated three times.

## Statical analysis

All statistical analyses were performed using SPSS software (IBM SPSS Statistics, version XX, Armonk, NY, USA). Data are presented as mean ± standard deviation (SD). Group comparisons were conducted using one-way analysis of variance (ANOVA) followed by Tukey's post hoc test to determine significant differences between treatment groups. A p-value < 0.05 was considered statistically significant. All experiments were independently repeated three times (n = 3) to ensure reproducibility.

## Results

### AC enhances p-ERK expression compared to BMP-2 and IGF in myoblasts

ERK activation is known to enhance the expression of tendon-specific markers such as Scleraxis and Tenomodulin, thereby improving mechanical properties. Treatment with AC significantly upregulated p-ERK protein expression in myoblasts compared to BMP-2 and IGF. At a concentration of 1 nM, there was no statistically significant difference in p-ERK expression among the treatment groups (Fig 1A). However, at 10 nM, p-ERK expression in the AC-treated group was significantly higher than in the BMP-2 and IGF-treated groups (p < 0.05; Fig 1B). These findings suggest that the effect of AC on protein upregulation is dose-dependent and may be more effective for TBI healing than individual growth factors.

### AC increases TnC expression relative to BMP-2, IGF, and VEGF

AC treatment also enhanced TnC expression in myoblasts compared to BMP-2, IGF, and VEGF. TnC upregulation has been associated with ECM remodeling and improved mechanical strength during tendon healing. Quantitative analysis of TnC protein revealed no significant difference among the groups at 1 nM (Fig 2A). However, at 10 nM, AC significantly increased TnC expression compared to BMP-2 (p < 0.01), IGF (p < 0.01), and VEGF (p < 0.05; Fig 2B). No significant difference was observed between AC and TGF-β1. These results suggest that AC may promote ECM remodeling more effectively than BMP-2, IGF, and VEGF.

### AC improves the Col I/III ratio compared to IGF

The Col I/III ratio serves as a critical molecular marker reflecting ECM organization, mechanical strength, and long-term functional recovery in tendon healing. Analysis of this ratio showed no significant differences at 1 nM among the groups (Fig 3A). However, at 10 nM, the AC-treated group exhibited a significantly higher Col I/III ratio compared to the IGF-treated group (p < 0.05; Fig 3B), with no significant differences observed when compared to the other growth factors. These results indicate a shift toward a more biomechanically favorable collagen ratio under AC treatment.

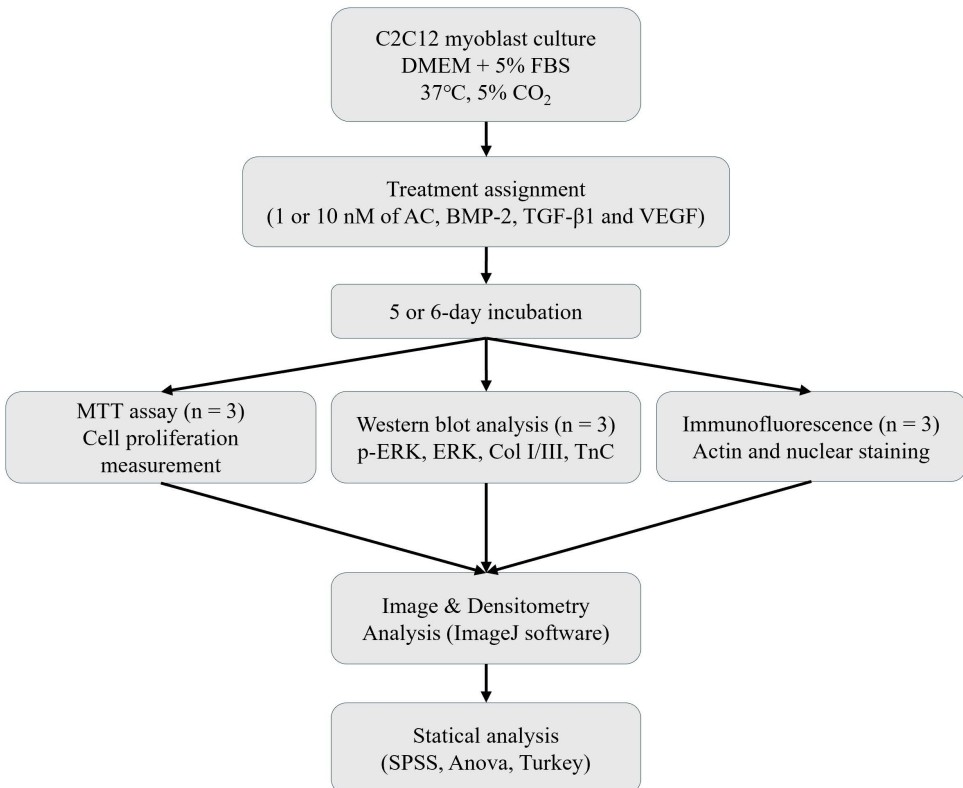

**Fig 1. Western blot analysis of p-ERK/ERK ratio in C2C12 myoblasts.** (A) At 1 nM, no significant differences were observed among treatment groups. (B) At 10 nM, AC significantly increased p-ERK/ERK ratio compared to BMP-2 and IGF-1. Data are presented as mean ± SD. * indicates p < 0.05.

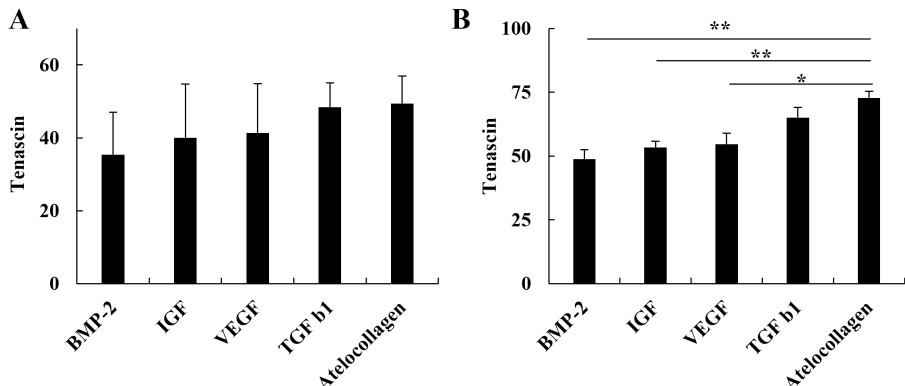

**Fig 2. Effect of AC and growth factors on TnC protein expression.** (A) At 1 nM, no significant difference in TnC expression was observed across groups. (B) At 10 nM, AC treatment significantly upregulated TnC compared to BMP-2, IGF-1, and VEGF, while no significant difference was noted with TGF-β1. Data are shown as mean ± SD. * indicates p < 0.05; ** indicates p < 0.01.

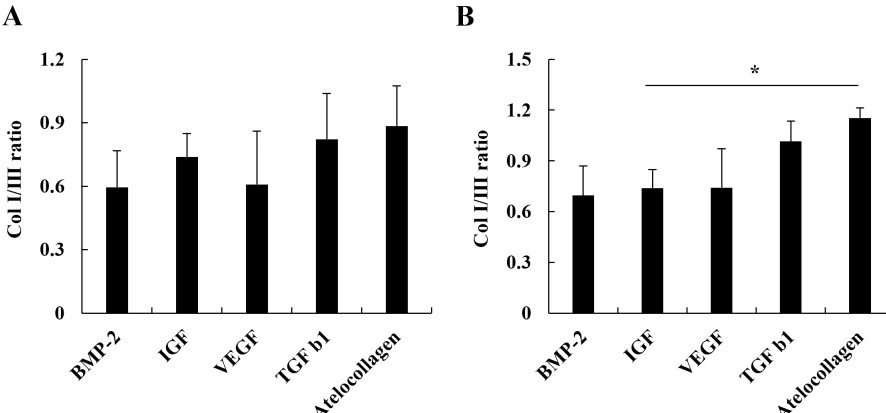

**Fig 3. Analysis of the Col I/III ratio in C2C12 cells treated with AC and each growth factor.** (A) No significant difference was observed among groups at 1 nM. (B) At 10 nM, the Col I/III ratio was significantly higher in the AC-treated group compared to IGF-1, suggesting enhanced ECM organization. Data represent mean ± SD. * indicates $p < 0.05$.

## AC enhances myoblast proliferation more effectively than VEGF and TGF-β1

Cell proliferation is a prerequisite for effective tendon healing, as it facilitates ECM regeneration and restoration of mechanical properties. In MTT assays, the AC-treated group exhibited higher proliferation rates than the VEGF-treated group (Fig 4). No significant difference was observed between AC and TGF-β1 (Table 1). These findings support the potential of AC to enhance cellular proliferation, which may contribute to improved tendon regeneration and biomechanical resilience.

## AC improves cell adhesion and structural integrity in vitro

Immunofluorescence staining supported these molecular findings, revealing greater actin filament organization in AC-treated cells compared to those treated with VEGF and TGF-β1 at 10 nM. Merged images of DAPI nuclear staining and actin cytoskeleton staining demonstrated more structured and aligned cytoskeletal architecture in the AC group, suggesting enhanced cell adhesion and structural integrity (Fig 5).

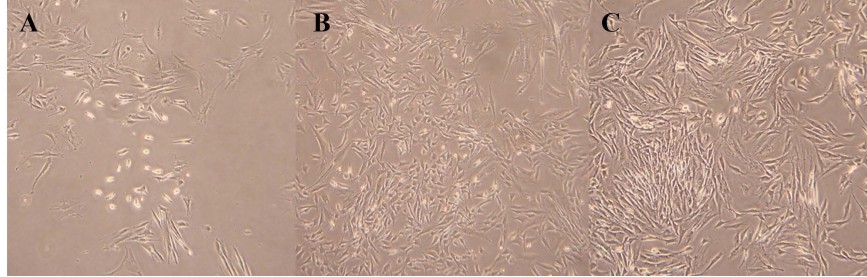

**Fig 4. Morphological assessment of C2C12 myoblasts treated with VEGF, TGF-β1, or AC (10 nM) for 6 days.** (A) VEGF group shows low confluence and limited cell-cell contact. (B) TGF-β1 group displays moderate density with partially aligned spindle morphology. (C) AC-treated cells demonstrate high confluence and uniform alignment, suggesting enhanced proliferation and cytoskeletal organization. All images captured under identical magnification.

**Table 1. Cell proliferation assay of C2C12 myoblasts treated with AC, VEGF, and TGF-β1.**

| | VEGFa | TGF-β1 | Atelocollagena |
|---|---|---|---|
| Average±SD | 39.96±3.71 | 47.95±2.90 | 57.30±7.65 |

Data are presented as mean±standard deviation (SD). a indicates p<0.05.

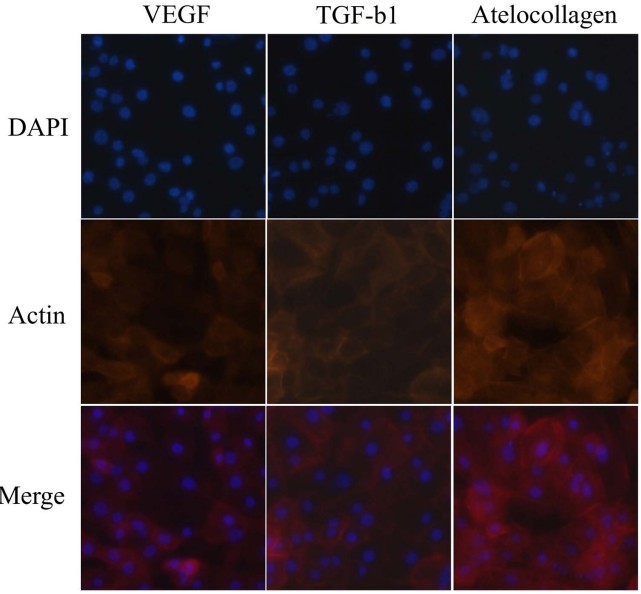

**Fig 5. Immunofluorescence images in C2C12 myoblasts following treatment with VEGF, TGF-β1, or AC at 10 nM for 6 days.** AC-treated cells exhibited enhanced F-actin filament organization with well-aligned, elongated stress fibers and increased cell spreading compared to VEGF- and TGF-β1-treated groups. In contrast, VEGF- and TGF-β1-treated cells showed a more disorganized actin network with less cytoplasmic extension. These findings suggest improved cytoskeletal integrity and cell adhesion under AC treatment. Red: phalloidin-stained F-actin; blue: DAPI-stained nuclei. Images were acquired using confocal microscopy under consistent acquisition settings.

## Discussions

The results of this study demonstrate that AC enhances key molecular markers associated with tendon healing, surpassing the effects of individual growth factors at a concentration of 10 nM. Specifically, AC significantly upregulated p-ERK and TnC expression, which are major regulators of cell proliferation and ECM remodeling, while also improving the Col I/III ratio, suggesting its ability to optimize tendon matrix composition. Furthermore, immunofluorescence analysis revealed enhanced cytoskeletal organization in AC-treated cells, further supporting its role in tenocyte function and adhesion. These findings strongly suggest that AC may serve as a superior biological augmentation strategy for tendon healing compared to growth factors.

ERK activation plays a key role in tendon healing by promoting tenocyte proliferation, ECM remodeling, and cytoskeletal organization [19]. Growth factors such as VEGF and TGF-β1 activate ERK signaling, leading to the upregulation of tendon-specific markers like Scleraxis and Tenomodulin, which are essential for ECM stabilization and functional tendon regeneration [20]. In a rotator cuff tear rat model, ERK activation has been shown to facilitate TBI healing by enhancing tenocyte proliferation and ECM remodeling, leading to improved collagen fiber organization and biomechanical strength [21]. Similarly, in a patellar tendon injury model, sustained ERK signaling promoted the formation of a well-structured

fibrocartilaginous transition zone, which is critical for restoring tendon-bone integration and preventing repair failure [22]. Furthermore, in an Achilles tendon rupture rat model, ERK-mediated upregulation of tenogenic markers such as Scleraxis and Tenomodulin contributed to enhanced ECM stabilization and mechanical properties of the healing tissue [23]. Our results demonstrate that AC significantly upregulated p-ERK protein levels compared to individual growth factors, suggesting that AC more effectively enhances tendon-specific cellular signaling and ECM integrity. This highlights AC's potential to activate key regenerative pathways more efficiently than conventional growth factor therapies.

TnC, a key ECM glycoprotein involved in tissue remodeling and mechanotransduction, is known to be regulated by ERK and TGF-β1 signaling and plays a crucial role in collagen fibrillogenesis, cellular migration, and mechanical stress adaptation [24]. In a study using C3H10T1/2 cells, BMP-12 treatment significantly upregulated TnC expression in a dose-dependent manner, with the highest expression observed at 10 ng/ml for 48 hours [25]. Additionally, fluorescence intensity analysis revealed a substantial increase in TnC expression compared to the non-induced control, indicating its pivotal role in tenogenic differentiation [26]. Clinically, elevated TnC expression has been linked to improved tendon healing, with studies reporting a correlation between its upregulation and enhanced ECM remodeling and mechanical strength of the repaired tissue [16]. In our study, AC-treated cells exhibited a significant increase in TnC protein expression, further supporting its role in collagen organization and TBI biomechanical strength enhancement. Given that TnC upregulation has been associated with superior biomechanical recovery in both preclinical and clinical models, these findings suggest that AC may provide a biologically favorable environment for tendon healing.

The regulation of the Col I/III ratio is critical for ECM remodeling and tendon biomechanical integrity [27]. During early tendon healing, increased Col III expression facilitates initial matrix deposition but is associated with scar formation and reduced mechanical strength [28]. As healing progresses, a transition toward Col I dominance promotes greater fibril organization, enhanced tensile strength, and improved functional recovery [29]. In a study using a rat Achilles tendon healing model, the regulation of the Col I/III ratio was shown to be a crucial determinant of biomechanical properties, with a Col I/III ratio shift correlating with a 40% increase in tensile strength after 8 weeks of healing [28]. In clinical trials, patients with a higher Col I/III ratio exhibited a 30–50% reduction in retear rates compared to those with an imbalanced collagen composition, emphasizing the importance of collagen composition in functional recovery [30]. In our study, AC-treated cells exhibited a significantly higher Col I/III ratio than those treated with individual growth factors. This suggests that AC not only promotes ECM remodeling but also facilitates the formation of a more biomechanically robust tendon matrix. Given that clinical studies have linked a higher Col I/III ratio to improved mechanical properties and reduced retear rates, these findings reinforce AC's potential as an effective biological augmentation strategy for tendon healing.

Immunofluorescence staining further corroborates these molecular findings, demonstrating enhanced actin filament organization in AC-treated cells, indicative of improved cytoskeletal alignment and cell adhesion [31]. Proper cytoskeletal organization is crucial for tenocyte function, as it facilitates cell-matrix interactions and enables mechanical loading in tendon adaptation, which is essential for maintaining tendon integrity under physiological stress [22,32]. ERK activation has been shown to modulate cytoskeletal dynamics by enhancing actin polymerization and focal adhesion assembly, thereby promoting cellular alignment and ECM stability [31]. Additionally, TnC, a key ECM glycoprotein involved in mechanical loading adaptation, was significantly upregulated in AC-treated cells, suggesting a more robust ECM remodeling response [31]. Our findings suggest that AC-mediated upregulation of these molecular pathways enhances cytoskeletal stability, ECM organization, and biomechanical resilience, supporting its potential as an effective biological augmentation strategy for tendon healing.

Although AC has been increasingly recognized for its regenerative potential in tendon healing, limited data exist regarding its dose-dependent effects on molecular signaling pathways. The present study provides compelling evidence that AC exhibits a dose-dependent upregulation of key molecular markers, including p-ERK, TnC, and the Col I/III ratio. Notably, treatment with 10 nM AC resulted in significantly enhanced activation of tendon-specific signaling and ECM remodeling compared to both 1 nM AC and individual growth factors. This finding suggests that a minimum threshold concentration is

required to sufficiently engage downstream signaling cascades, particularly those involved in cellular proliferation, ECM organization, and biomechanical reinforcement. Similar dose-dependent effects have been observed with growth factor-based therapies, where insufficient concentrations fail to induce tenogenic differentiation due to limited receptor activation or inefficient intracellular signal transduction [15,31]. These results highlight the importance of optimizing AC dosage in clinical applications and support the use of higher-concentration formulations to achieve more consistent and robust biological augmentation of tendon healing.

The results of this study underscore several distinct advantages of AC as a biomaterial for tendon repair. Unlike exogenous growth factors that are rapidly degraded in vivo and therefore require encapsulation or repeated administration, AC exhibits sustained biochemical stability, enabling prolonged bioactivity without the need for complex delivery systems [33]. Our findings demonstrate that AC promotes tendon regeneration through a multifaceted mechanism involving sustained ERK activation, VEGF-mediated neovascularization, upregulation of TnC, and optimization of the Col I/III ratio, suggesting its superiority over individual growth factors. Clinically, these properties of AC may contribute to improved biomechanical healing, reduced risk of retear, and serve as a cost-effective alternative to growth factor-based therapies [16,30,34]. Therefore, AC represents a promising biological augmentation strategy that addresses both molecular and mechanical aspects of tendon healing.

Despite these promising results, this study has several limitations. First, the experiments were conducted in an in vitro model, which does not fully replicate the complex biomechanical and biochemical environment of in vivo tendon healing. Second, only a single concentration (10nM) was evaluated in depth, whereas a dose-dependent study in an in vivo setting may provide additional insights into AC's therapeutic potential. Furthermore, long-term studies assessing mechanical strength and functional outcomes in tendon repair models are necessary to confirm the translational applicability of AC in clinical settings.

## Conclusions

AC significantly enhanced the protein expression of key molecular markers associated with tenocyte proliferation, ECM remodeling, and tendon structural integrity, demonstrating a more pronounced effect than individual growth factors. By modulating critical molecular pathways involved in tendon healing, AC may serve as a more effective biological tendon augmentation strategy than growth factor therapy, providing a superior approach for improving both histological and biomechanical TBI healing following tendon repair.

## Supporting information

**S1 File. Raw data.** Figure 2–4 and Table 1. Raw data for Figs 2–4 and Table 1. This file contains the original datasets used in the present study, including quantitative tenascin-C (TnC) protein expression data at 1nM and 10nM for atelocollagen (AC) and individual growth factors with corresponding statistical comparisons (Fig 2); quantitative analysis of the collagen type I/III ratio in C2C12 cells treated with AC and each growth factor at both 1nM and 10nM concentrations (Fig 3); morphological assessment data from C2C12 myoblasts treated with VEGF, TGF-β1, or AC (10nM), including confluence, spindle morphology, and alignment scoring (Fig 4); and MTT cell proliferation assay measurements for C2C12 myoblasts treated with AC, VEGF, and TGF-β1, including mean, standard deviation, and statistical significance (Table 1). Data are organized by figure/table number, experimental group, treatment concentration, and replicate number. (XLSX)

## Author contributions

**Data curation:** Sung-Jin Park, Seok Won Chung.

**Formal analysis:** Jong Pil Yoon, Kyung-Soo Oh.

**Funding acquisition:** Seok Won Chung.

**Methodology:** Sung-Jin Park, Seok Won Chung.

**Resources:** Sung-Jin Park, Jong Pil Yoon, Kyung-Soo Oh, Seok Won Chung.

**Writing – original draft:** Sung-Jin Park.

**Writing – review & editing:** Sung-Jin Park, Seok Won Chung.

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
