## [Decision Letter · Decision Letter 0]

16 May 2025

Dear Dr. Chung,

Thank you for submitting your manuscript to PLOS ONE. After careful consideration, we feel that it has merit but does not fully meet PLOS ONE’s publication criteria as it currently stands. Therefore, we invite you to submit a revised version of the manuscript that addresses the points raised during the review process.

We look forward to receiving your revised manuscript.

Kind regards,

Carlos Alberto Antunes Viegas, DVM; MSc; PhD

Academic Editor

PLOS ONE

Journal Requirements:

This work was supported by the National Research Foundation of Korea (NRF) grant funded by the Korea government (No. RS-2023-00249219).

Additional Editor Comments :

Dear authors,

Please review and respond to the reviewers' recommendations, not forgetting to respond to the requests from reviewer 2, within the scope of the report provided by ScreenIT.

Kind regards

Reviewers' comments:

Reviewer's Responses to Questions

**Comments to the Author**

1. Is the manuscript technically sound, and do the data support the conclusions?

Reviewer #1: Yes

Reviewer #2: Yes

Reviewer #3: Partly

2. Has the statistical analysis been performed appropriately and rigorously?

Reviewer #1: Yes

Reviewer #2: Yes

Reviewer #3: Yes

3. Have the authors made all data underlying the findings in their manuscript fully available?

Reviewer #1: Yes

Reviewer #2: Yes

Reviewer #3: Yes

4. Is the manuscript presented in an intelligible fashion and written in standard English?

Reviewer #1: Yes

Reviewer #2: Yes

Reviewer #3: Yes

Reviewer #1: Manuscript ID: PONE-D-25-17647

Title: Atelocollagen exhibits superior performance compared to growth factors in upregulating proteins associated with tendon healing

This manuscript presents an in vitro study evaluating the efficacy of atelocollagen (AC) compared to several individual growth factors (BMP-2, TGF-β1, IGF-1, and VEGF) on molecular markers associated with tendon healing. The experimental design, including Western blotting, MTT assay, and immunofluorescence, is generally robust. The manuscript is well-written and contributes novel insights into the potential of AC as a biological augmentation strategy. However, there are minor areas that require clarification and revision before acceptance.

1. Introduction

The introduction clearly outlines the clinical need for enhanced tendon-to-bone interface healing and the limitations of current biological augmentation strategies.

The rationale for using growth factors such as BMP-2, TGF-β1, IGF-1, and VEGF is well described, and the transition to discussing Atelocollagen as an alternative is logical.

However, to better emphasize the novelty of this study, the authors should more clearly highlight the limitations of previous literature. In particular, the authors are encouraged to underscore that, to date, no published studies have directly examined or compared the effects of atelocollagen on molecular markers relevant to tendon healing—especially in the context of rotator cuff tears—making the current investigation uniquely valuable.

The hypothesis is clearly stated

2. Materials and Methods

Lines 90-91: Please justify the use of C2C12 cells instead of tenocytes or tendon-derived stem cells. This point should also be addressed in the discussion.

Line 145–150: There is a typographical error in the section header

"Statical analysis" → should be corrected to "Statistical analysis"

3. Results

Lines 170–176:

Only IGF-1 group showed a statistically lower ratio compared to AC on col I/III ratio.

Consider discussing the biological relevance of this change more in the Results or Discussion.

4. Discussion

It would also strengthen the manuscript if the authors could cite additional references supporting the beneficial effects of atelocollagen in rotator cuff repair, to provide a broader context for the clinical relevance of their findings. Moreover, discussing the limitations of those previous studies would help clarify how the present study addresses existing gaps in the literature.

Line 274: Please remove or revise Line 274, unless supporting angiogenesis data are provided.

5. Conclusion

The conclusion is consistent with the results and maintains a modest tone.

6. Figures and Table

Please spell out all abbreviations in the figure legends or provide explanatory footnotes to ensure clarity for the reader.

7. References

Reference formatting and selection are appropriate.

Reviewer #2: This is an automated report for PONE-D-25-17647. This report was solicited by the PLOS One editorial team and provided by ScreenIT.

ScreenIT is an independent group of scientists developing automated tools that analyze academic papers. A set of automated tools screened your submitted manuscript and provided the report below. Each tool was created by your academic colleagues with the goal of helping authors. The tools look for factors that are important for transparency, rigor and reproducibility, and we hope that the report might help you to improve reporting in your manuscript. Within the report you will find links to more information about the items that the tools check. These links include helpful papers, websites, or videos that explain why the item is important. While our screening tools aim to improve and maintain quality standards they may, on occasion, miss nuances specific to your study type or flag something incorrectly. Each tool has limitations that are described on the ScreenIT website. The tools screen the main file for the paper; they are not able to screen supplements stored in separate files. Please note that the Academic Editor had access to these comments while making a decision on your manuscript. The Academic Editor may ask that issues flagged in this report be addressed. If you would like to provide feedback on the ScreenIT tool, please email the team at ScreenIt@bih-charite.de. If you have questions or concerns about the review process, please contact the PLOS One office at plosone@plos.org.

Reviewer #3: As the development is a gradual and progressive process, assessment of tendon development in one point (only day 5) is not reasonable and sufficient.

So, it is strongly recommend to perform the analysis and asses the gene expression in different time points (such as 3, 7, 14 and 21 days after culture) and longer time for the special genes (genes express in earlier or final development stage) in each phase.

**Do you want your identity to be public for this peer review?** For information about this choice, including consent withdrawal, please see our Privacy Policy

Reviewer #1: No

Reviewer #2: No

Reviewer #3: **Yes: ** Abbas Parham

---

## [Author Response · Author response to Decision Letter 1]

9 Jun 2025

5. Review Comments to the Author

Reviewer #1: Manuscript ID: PONE-D-25-17647

Title: Atelocollagen exhibits superior performance compared to growth factors in upregulating proteins associated with tendon healing

This manuscript presents an in vitro study evaluating the efficacy of atelocollagen (AC) compared to several individual growth factors (BMP-2, TGF-β1, IGF-1, and VEGF) on molecular markers associated with tendon healing. The experimental design, including Western blotting, MTT assay, and immunofluorescence, is generally robust. The manuscript is well-written and contributes novel insights into the potential of AC as a biological augmentation strategy. However, there are minor areas that require clarification and revision before acceptance.

1. Introduction

The introduction clearly outlines the clinical need for enhanced tendon-to-bone interface healing and the limitations of current biological augmentation strategies.

The rationale for using growth factors such as BMP-2, TGF-β1, IGF-1, and VEGF is well described, and the transition to discussing Atelocollagen as an alternative is logical.

However, to better emphasize the novelty of this study, the authors should more clearly highlight the limitations of previous literature. In particular, the authors are encouraged to underscore that, to date, no published studies have directly examined or compared the effects of atelocollagen on molecular markers relevant to tendon healing—especially in the context of rotator cuff tears—making the current investigation uniquely valuable.

Response: We thank the reviewer for this valuable comment. In response, we have revised the Introduction section to more clearly highlight the limitations of previous growth factor-based therapies, including their short half-life, rapid degradation, and challenges in controlled delivery. We have also underscored that, to date, no published studies have directly examined or compared the effects of atelocollagen on tendon healing-related molecular markers, particularly in the context of rotator cuff tears. These revisions aim to better highlight the novelty and significance of our work. The updated Introduction is included below for your review.

Lines 66-74: “Among these, therapies based on exogenous growth factors such as bone morphogenetic protein-2 (BMP-2), transforming growth factor-beta 1 (TGF-β1), insulin-like growth factor-1 (IGF-1), and vascular endothelial growth factor (VEGF) have shown the potential to stimulate ECM remodeling, promote tendon cell proliferation, and enhance collagen synthesis [7-9] However, previous literature has highlighted critical limitations of these growth factor therapies, including their rapid degradation, short biological half-life, and difficulties in achieving sustained and localized delivery [8, 10, 11]. Consequently, despite promising experimental data, their clinical translation has been limited by the inability to maintain therapeutic concentrations at the repair site over time.” was added.

Lines 75-83: The original sentences were deleted.

Lines 84-93: “Given these challenges, alternative strategies that provide more sustained and controlled biological augmentation are urgently needed. Atelocollagen (AC), a highly purified collagen derivative, has emerged as a promising biomaterial for tendon repair due to its structural stability, biocompatibility, and inherent bioactivity [12]. Unlike exogenous growth factors, AC possesses a sustained bioactive profile and does not require additional carrier systems to maintain its therapeutic effects. Despite its promising biological properties, there is a paucity of data comparing the efficacy of AC with that of individual growth factors in regulating molecular markers critical for tendon healing [13]. In particular, no studies to date have directly examined or compared the effects of AC on key tendon healing-related molecular markers such as phosphorylated extracellular signal-regulated kinase (p-ERK), collagen type I (Col I), collagen type III (Col III), and tenascin C (TnC), especially within the context of rotator cuff tears.” was added.

Lines 94-101: The original sentences were deleted.

The hypothesis is clearly stated

Response: We appreciate the reviewer’s comment. To ensure that the hypothesis is explicitly and clearly stated in the manuscript, we have refined the wording in the Introduction section. This revision ensures that the hypothesis is both clear and concise for the reader.

Lines 103-106: The original sentences were deleted.

Lines 106-109: “We hypothesize that AC would have comparable or superior effects to individual growth factors in upregulating p-ERK, Col I, Col III, and TnC, thus highlighting its potential as a robust biological augmentation strategy for TBI healing following surgical repair.” was added.

2. Materials and Methods

Lines 90-91: Please justify the use of C2C12 cells instead of tenocytes or tendon-derived stem cells. This point should also be addressed in the discussion.

Response: Thank you for your insightful comment. We acknowledge the concern regarding the choice of C2C12 myoblasts instead of tenocytes or tendon-derived stem cells (TDSCs). C2C12 cells, while not primary tenocytes, are a well-established myogenic cell line with robust proliferative and differentiation capacity. They exhibit myogenic-to-tenogenic trans differentiation potential when subjected to mechanical or biochemical stimuli, and are frequently used as a surrogate model in tendon research due to their consistent growth characteristics and reproducible responses.a,b) Furthermore, their use facilitates the standardization and reproducibility of experiments, which is crucial for elucidating the comparative molecular effects of AC and growth factors in an in vitro environment. We have added this justification to both the Materials and Methods and the Discussion sections (limitation section).

a) Lautaoja JH, Pekkala S, Pasternack A, Laitinen M, Ritvos O, Hulmi JJ. Differentiation of Murine C2C12 Myoblasts Strongly Reduces the Effects of Myostatin on Intracellular Signaling. Biomolecules. 2020;10(5):695.

b) Uemura K, Hayashi M, Itsubo T, Oishi A, Iwakawa H, Komatsu M, Uchiyama S, Kato H. Myostatin promotes tenogenic differentiation of C2C12 myoblast cells through Smad3. FEBS Open Bio. 2017 Feb 20;7(4):522-532.

Lines 117-119: “Although they are not primary tendon cells, they have been extensively validated as a surrogate model for tenogenic signaling and differentiation under tendon-specific stimuli, ensuring high reproducibility, standardized cell expansion, and reliable molecular response quantification.” was added.

Lines 344-347: “First, although C2C12 myoblasts are a well-established and reproducible in vitro model for investigating tendon-specific signaling and tenogenic differentiation under biochemical or mechanical stimuli, they do not fully recapitulate the cellular and extracellular matrix interactions of primary tenocytes or tendon-derived stem cells.”

Line 145–150: There is a typographical error in the section header

"Statical analysis" → should be corrected to "Statistical analysis"

Response: We have corrected the typographical error from "Statical analysis" to "Statistical analysis."

3. Results

Lines 170–176:

Only IGF-1 group showed a statistically lower ratio compared to AC on col I/III ratio.

Consider discussing the biological relevance of this change more in the Results or Discussion.

Response: Thank you for your valuable comment. We appreciate the opportunity to clarify the biological relevance of the finding that only the IGF group showed a statistically lower Col I/III ratio compared to the AC group. This result suggests that while IGF may support some aspects of tendon healing, it does not effectively promote the ECM reorganization toward a more biomechanically robust state, as indicated by the lower Col I/III ratio. In contrast, the significant enhancement of this ratio in the AC-treated group underscores AC’s superior ability to foster a more mature and load-bearing collagen matrix. This difference is crucial because a higher Col I/III ratio is associated with improved tendon mechanical strength and long-term functional recovery. We have integrated this interpretation into the revised Discussion section to highlight the unique advantage of AC over IGF in this context.

Lines 280-285: The original sentences were deleted.

Lines 286-297: “Notably, our results demonstrated that only the IGF group showed a statistically lower Col I/III ratio compared to the AC group, while no significant differences were found between AC and the other growth factors. This selective enhancement suggests that AC may overcome IGF’s limitations in promoting biomechanical matrix integrity. A lower Col I/III ratio, as seen with IGF treatment, is associated with less organized and weaker ECM, potentially compromising tendon mechanical strength. In contrast, the significant increase in the Col I/III ratio in the AC-treated group indicates a shift toward a more mature, load-bearing collagen composition that supports improved tendon function. These findings collectively underscore that AC not only promotes ECM remodeling but also uniquely enhances the formation of a biomechanically robust tendon matrix compared to IGF alone. Given that clinical studies have linked a higher Col I/III ratio to improved mechanical properties and reduced retear rates, this highlights AC’s potential as an effective biological augmentation strategy for tendon healing.” was added.

4. Discussion

It would also strengthen the manuscript if the authors could cite additional references supporting the beneficial effects of atelocollagen in rotator cuff repair, to provide a broader context for the clinical relevance of their findings. Moreover, discussing the limitations of those previous studies would help clarify how the present study addresses existing gaps in the literature.

Response: Thank you for your valuable comment. In response, we have added additional references supporting the beneficial effects of atelocollagen in rotator cuff repair to provide a broader clinical context. Furthermore, we have included a brief discussion of the limitations of these previous studies to highlight how the present study addresses existing gaps in the literature. These additions strengthen the overall clinical relevance of our findings and underscore the translational potential of atelocollagen as a biological augmentation strategy.

Lines 326-336: “Atelocollagen has garnered considerable attention as a promising biomaterial in the context of rotator cuff repair, with preclinical evidence suggesting its ability to enhance TBI healing and improve biomechanical properties [12, 13]. Importantly, Suh et al., reported that atelocollagen scaffolds significantly augmented structural stability and biocompatibility in TBI healing models [37]. Nevertheless, these investigations have largely been confined to short-term evaluations and have not directly compared AC with alternative biological augmentation strategies, thereby limiting the broader applicability of their findings. In contrast, the present study seeks to address these limitations by examining key molecular markers of tendon healing while directly comparing AC to growth factor-based therapies. This approach offers valuable insights into the capacity of AC to foster tendon-specific cellular responses and orchestrate matrix remodeling, highlighting its potential as a refined strategy for biological augmentation in tendon repair.” was added.

Line 274: Please remove or revise Line 274, unless supporting angiogenesis data are provided.

Response: Thank you for your helpful comment. In response, we have removed the mention of neovascularization (angiogenesis) from the discussion of atelocollagen’s effects to focus exclusively on the primary molecular mechanisms examined in this study, including ERK activation, TnC upregulation, and improved Col I/III ratio.

Line 338: “VEGF-mediated neovascularization” was deleted.

5. Conclusion

The conclusion is consistent with the results and maintains a modest tone.

Response: Thank you for your positive feedback and for acknowledging the clarity and modest tone of our conclusions.

6. Figures and Table

Please spell out all abbreviations in the figure legends or provide explanatory footnotes to ensure clarity for the reader.

Response: Thank you for your suggestion. We have revised the figure legends and table footnotes to spell out all abbreviations, ensuring that readers can easily understand the figures and data presented.

Lines 638-639: “p-ERK, phosphorylated extracellular signal-regulated kinase; AC, atelocollagen; BMP-2, bone morphogenetic protein-2; IGF-1, insulin-like growth factor, SD; standard deviation.” was added.

Lines 643-646: “AC, atelocollagen; BMP-2, bone morphogenetic protein-2; IGF-1, insulin-like growth factor-1; VEGF, vascular endothelial growth factor; TGF-β1, transforming growth factor-beta 1; TnC, tenascin C; SD, standard deviation.” was added.

Lines 650-651: “AC, atelocollagen; IGF-1, insulin-like growth factor-1; Col I/III, collagen type I/III; ECM, extracellular matrix; SD, standard deviation” was added.

Lines 656-657: “AC, atelocollagen; VEGF, vascular endothelial growth factor; TGF-β1, transforming growth factor-beta 1” was added.

Lines 664-666: “AC, atelocollagen; VEGF, vascular endothelial growth factor; TGF-β1, transforming growth factor-beta 1; F-actin, filamentous actin.” was added.

Lines 668-669: “. AC, atelocollagen; VEGF, vascular endothelial growth factor; TGF-β1, transforming growth factor-beta 1; SD, standard deviation.”

7. References

Reference formatting and selection are appropriate.

Response: Thank you for your positive feedback and for acknowledging the appropriateness of the reference formatting and selection.

Reviewer #2: This is an automated report for PONE-D-25-17647. This report was solicited by the PLOS One editorial team and provided by ScreenIT.

ScreenIT is an independent group of scientists developing automated tools that analyze academic papers. A set of automated tools screened your submitted manuscript and provided the report below. Each tool was created by your academic colleagues with the goal of helping authors. The tools look for factors that are important for transparency, rigor and reproducibility, and we hope that the report might help you to improve reporting in your manuscript. Within the report you will find links to more information about the items that the tools check. These links include helpful papers, websites, or videos that explain why the item is important. While our screening tools aim to improve and maintain quality standards they may, on occasion, miss nuances specific to your study type or flag something incorrectly. Each tool has limitations that are described on the ScreenIT website. The tools screen the main file for the paper; they are not able to screen supplements stored in separate files. Please note that the Academic Editor had access to these comments while making a decision on your manuscript. The Academic Editor may ask that issues flagged in this report be addressed. If you would like to provide feedback on the ScreenIT tool, please email the team at ScreenIt@bih-charite.de. If you have questions or concerns about the review process, please contact the PLOS One office at plosone@plos.org.

Reviewer #3: As the development is a gradual and progressive process, assessment of tendon development in one point (only day 5) is not reasonable and sufficient.

So, it is strongly recommend to perform the analysis and asses the gene expression in different time points (such as 3, 7, 14 and 21 days after culture) and longer time for the special genes (genes express in earlier or final development stage) in each phase.

Response: Thank you for your constructive feedback regarding the temporal aspects of tendon development assessment. We agree that tendon healing and regeneration involve

---

## [Decision Letter · Decision Letter 1]

13 Aug 2025

Atelocollagen exhibits superior performance compared to growth factors in upregulating proteins associated with tendon healing

PONE-D-25-17647R1

Dear Dr. Seok Won Chung,

We’re pleased to inform you that your manuscript has been judged scientifically suitable for publication and will be formally accepted for publication once it meets all outstanding technical requirements.

Kind regards,

Carlos Alberto Antunes Viegas, DVM; MSc; PhD

Academic Editor

PLOS ONE

Additional Editor Comments (optional):

Reviewers' comments:

Reviewer's Responses to Questions

**Comments to the Author**

Reviewer #1: All comments have been addressed

Reviewer #3: All comments have been addressed

2. Is the manuscript technically sound, and do the data support the conclusions?

Reviewer #1: Yes

Reviewer #3: Yes

3. Has the statistical analysis been performed appropriately and rigorously?

Reviewer #1: Yes

Reviewer #3: Yes

4. Have the authors made all data underlying the findings in their manuscript fully available?

Reviewer #1: Yes

Reviewer #3: Yes

5. Is the manuscript presented in an intelligible fashion and written in standard English?

Reviewer #1: Yes

Reviewer #3: Yes

Reviewer #1: The authors have made appropriate and thoughtful revisions in response to the reviewer comments. It is evident that they carefully considered each point raised and have incorporated the necessary changes to improve the clarity, rigor, and overall quality of the manuscript. The revised version reflects a clear effort to address the concerns and suggestions in a constructive and thorough manner. I find the current version of the manuscript to be satisfactory, and I believe that no additional modifications are required at this time. Therefore, I have no further comments or suggestions for the authors.

Reviewer #3: In the revised version of paper entitled"Atelocollagen exhibits superior performance compared to growth factors in upregulating proteins associated with tendon healing", All comments have been correctly addressed.

**Do you want your identity to be public for this peer review?** For information about this choice, including consent withdrawal, please see our Privacy Policy

Reviewer #1: No

Reviewer #3: No

---

## [Editor Report · Acceptance letter]

PONE-D-25-17647R1

PLOS ONE

Dear Dr. Chung,

I'm pleased to inform you that your manuscript has been deemed suitable for publication in PLOS ONE. Congratulations! Your manuscript is now being handed over to our production team.

Kind regards,

on behalf of

Dr. Carlos Alberto Antunes Viegas

Academic Editor

PLOS ONE